# Complex Substrate Leading to PVC-Mediated Systolic Dysfunction in addition to Sustained Monomorphic VT in Repaired Tetralogy of Fallot

**DOI:** 10.3390/diagnostics14020158

**Published:** 2024-01-10

**Authors:** Cosmin Cojocaru, Silvia Deaconu, Viviana Gondos, Sebastian Onciul, Ioana Petre, Oana Gheorghe-Fronea, Radu Vătășescu

**Affiliations:** 1Faculty of Medicine, “Carol Davila” University of Medicine and Pharmacy, 050474 Bucharest, Romania; cosmin.cojocaru@drd.umfcd.ro (C.C.); sebastian.onciul@gmail.com (S.O.); i_comanescu@yahoo.com (I.P.); dr.fronea79@gmail.com (O.G.-F.); 2Cardiology Department, Emergency Clinical Hospital of Bucharest, 014461 Bucharest, Romania; 3ARES Centers, 021967 Bucharest, Romania; si.deaconu@gmail.com; 4Department of Medical Electronics and Informatics, Polytechnic University, 060042 Bucharest, Romania; viviana.gondos@hellimed.ro

**Keywords:** repaired Tetralogy of Fallot, ventricular tachycardia, ablation, LV systolic dysfunction

## Abstract

Frequent premature ventricular complexes (PVCs) are associated with deleterious effects on left ventricular (LV) function in various clinical scenarios. Repaired tetralogy of Fallot (rTOF) is frequently affected by sustained ventricular arrhythmias dependent on complex post-surgical substrates. However, there is limited data regarding the potential of arrhythmogenic isthmuses to generate frequent PVCs and PVC-mediated LV systolic dysfunction development in rTOF. We present a case of rTOF experiencing relatively infrequent episodes of internal shocks for episodes of sustained monomorphic ventricular tachycardia and a high burden of PVCs associated with left ventricular systolic function deterioration, in which the successful substrate ablation of the anatomical VT isthmuses also led to PVC abolition and consequently to LV systolic function normalization. In such cases, understanding the pathogenic mechanisms that lead to LV dysfunction is only possible by rigorous clinical reasoning, which leads to a tailored specific treatment.

Repaired Tetralogy of Fallot (rTOF) patients may experience episodes of ventricular tachycardia (VT) which are most commonly dependent on anatomical and post-surgical isthmuses [1]. Progressive right ventricle (RV) dilation may further create additional ventricular arrhythmic substrate. Left ventricular (LV) cardiomyopathic changes have recently been described in rTOF patients and influence long-term prognosis [2,3]. However, in the presence of high burdens of premature ventricular contractions (PVCs), mild LV systolic dysfunction in such settings can be difficult to interpret. We describe the case of an rTOF patient with frequent PVCs, mild LV systolic dysfunction and intermittent episodes of sustained monomorphic VT treated by internal shocks who underwent catheter ablation. Due to its post-procedural resolution, we considered the LV systolic dysfunction to have been induced by PVCs.

This is the case of a 43-year-old male patient who was referred for episodes of palpitations and presyncope. The patient had undergone surgical correction of Tetralogy of Fallot (TOF) with trans-atrial–trans-annular pulmonary patch (TAP) at age four and had been compliant with treatment and annual reevaluations. Catheter ablation had been recently attempted without success in another center for repetitive episodes of scar-dependent monomorphic VT and frequent monomorphic right bundle branch (RBBB)-like PVCs (which had been diagnosed 3 years prior to this referral. Figure 1A depicts the baseline sinus rhythm with RBBB-QRS of 170 msec and isolated PVC RBBB morphology with superior axis. Figure 1B,C show the intraprocedural VTs eliminated during the initial catheter ablation procedure with LBBB morphology and superior axis with ventriculoatrial dissociation. Programmed electrical stimulation (PES) protocol (paced beats are marked by the horizontal arrow) performed at the end of the previous (first) procedure induced a residual RBBB VT with inferior axis (Figure 1D).

Therefore, a dual-chamber implantable cardioverter defibrillator (ICD) was implanted. Considering the patient had experienced two adequate shocks in the course of three months, we decided to perform a staged ablation (RV substrate followed by (apparently) LV-originating PVCs). The patient had no other relevant comorbidities. His medication included Nebivolol 5 mg QD, Furosemide 40 mg QD, Spironolactone 25 mg QD and antiarrhythmic treatment (AAD) with Amiodarone 200 mg QD. SGLT-2 inhibitors had not been prescribed as there was no formal indication at that time for patients with a mildly reduced LV ejection fraction (HFmrEF). Pathogenic medication targeting the renin-angiotensin-aldosterone system was not administered due to drug-induced hypotension (and modest guideline-based indications and evidence [4]). There were mild signs of peripheral edema evident at admission, and thus, Furosemide was transiently administered in intravenous higher doses during the 5-day admission to optimize systemic congestion control. The only available NT-proBNP dosing at the time of admission was 1863 pg/mL. There was no thyroid dysfunction or other reversible factors that might have facilitated the appearance of VTs or PVCs. Repaired TOF is the epitome of surgically created anatomical isthmuses (AI) (which create unexcitable conduction barriers) accompanied by areas of slow conduction (especially in the context of progressive interstitial fibrosis and dilation [5,6]). This is why most (>80%) arrhythmic episodes in primary and secondary prevention ICD rTOF patients are sustained monomorphic VTs and are the leading cause of sudden cardiac death [7]. Sotalol was considered as an antiarrhythmic alternative after Amiodarone failure [5]. However, motivated by both patient preference and the expected well-defined typical rTOF myocardial substrate, we considered ablation therapy to have a higher potential long-term benefit compared to an AAD-based strategy. In this sense, recent European guidelines clarify how catheter or surgical-based substrate ablation (class IIa), electrophysiologic study (class IIa) and ICD (IIa for primary prevention) are to be implemented into rTOF management [8]. Ablation, however, can be challenging due to intracardiac prosthetic material (sutures, patches, grafts, conduits, etc.), calcifications, hypertrophied myocardium and displaced anatomical structures [5]. Preprocedural imaging is therefore highly valuable for tridimensional anatomy reconstruction (especially by high-resolution cardiac computed tomography (CT)) and tissue characterization (cardiac magnetic resonance (CMR)). Due to recurrent ICD shocks despite AAD and presumed PVC-induced LV systolic dysfunction, we planned a staged ablation strategy of AI ablation followed by PVC origin mapping. The particular effect of PVC abolition after right-sided ablation is important to be clarified. Figure 1D shows the residual VT which was still inducible after the first ablation attempt prior to referral and Figure 1A depicts the high-burden PVC morphology. Recent publications have reiterated the specific morphologies for typical idiopathic RV and LV VT origins [9,10,11]. These, however, have been validated in structurally normal hearts. Based on these characteristics, both the residual VT and the frequent PVC exhibited morphologies potentially suggestive of left-sided origins. The VT’s inferior axis and positive precordial concordance could have initially been interpreted as an anterolateral mitral annulus exit point. This is why understanding the rTOF substrate is critical, as recently shown by Brouwer et al. [6]. In fact, in rTOF, RBBB-type VTs encompass a third of all anatomical isthmus-dependent VTs and are mostly targetable by right-sided approaches [6]. This pattern usually results from the anatomical displacement of the infundibular septum and the clockwise activation of an anatomical isthmus (AI) between the pulmonic valve and the septal patch (typically considered the “third” type of rTOF isthmus; the narrowest and most arrhythmogenic of all [12]) which matches the residual VT’s morphology [6]. Figure 2 and Figure 3 (see below) show the significantly clockwise-shifted anatomical positions of the RV (anteriorized) and LV (posteriorized). Furthermore, the superior axis with positive QRS in lead D1, positive QRS in V1-V4 and R amplitude similar to S in lead V5 could have indicated a posterior fascicular/papillary muscle origin (Figure 1A). However, as recently described, myocardial activation via the tricuspid-to-septal-patch (AI-4) anatomical isthmus can result in a cranially oriented pattern with a more leftward exit point compared to AI-3 morphologies and create the described pattern of activation. Having achieved the non-inducibility of monomorphic VTs at final PES portends a favorable prognosis with regard to VT recurrence and VT-related mortality in this case. Existing data show that VT abolition by ablation is associated with low long-term mortality and provides the possibility to discontinue AADs. However, repeat ablations can be required in up to 17.5% due to recurrences [13,14,15].

A transthoracic echocardiography showed RV dilation (diastolic area 60.8 sqcm, 64 mm basal diameter) and mild systolic dysfunction (FACVD 32%, TAPSE 17 mm, S’t 11 cm/s) (Figure 2A–C). The single-coil RV defibrillation lead is also visible in Figure 2A. Figure 2B demonstrates a paradoxical movement of the interventricular septum due to RV overload and severe pulmonary regurgitation (PR). Severe PR and mild pulmonary stenosis (Figure 2D) (PHT 123 ms, regurgitant jet > ⅔ of RVOT area, vena contracta 9 mm, EROA 70 mm^2^) were previously noted for which the patient was scheduled for transcatheter pulmonary valve replacement (PVR), considering the past episodes of systemic congestion. Moderate-to-severe functional tricuspid regurgitation with significant enlargement of the tricuspid annulus (53 mm) (Doppler CW tracing in Figure 2E). The LV was dilated (89 mL/sqm) and mildly diffusely hypokinetic with a biplane Simpson LVEF of 48%. There was no LV dysfunction prior to the occurrence of the PVCs. There was no residual shunt visible. A previous cardiac magnetic resonance (CMR) scan documented minimal fibrosis at the border of the right ventricular outflow tract (RVOT) and septal patches and severe RV dilation (209 mL/sqm). Coronary computed tomography angiography (CTPA) showed no significant coronary lesions.

Ablation was performed at the site of the isthmus located between the septal patch and the pulmonary annulus (yellow arrow in Figure 3C–E) and between the septal patch and the tricuspid annulus (green arrow in Figure 3D,E) by using remote magnetic navigation. Notably, there was concomitant complete PVC abolition after ablation between the septal patch and the tricuspid annulus (green arrow); hence, the LV was no longer instrumented. An end-procedural aggressive (up to four extra stimuli) programmed ventricular stimulation induced only a very short episode of non-sustained polymorphic VT. During the follow-up, a seven-day Holter monitoring 1-month post-ablation revealed less than 0.5% asymptomatic PVCs with distinct morphology. An ICD interrogation showed no recurrent VT episodes. At 1 month follow-up, echocardiography showed LVEF normalization (biplane Simpson 57%) and NT-proBNP values decreased to 642 pg/mL. There was no change, however, in the RV function or dimensions.

Understanding the underlying cause of LV dysfunction is a further critical aspect. LV dysfunction development in rTOF patients is significantly more frequent than formerly estimated (up to 23.6–49% in CMR studies [2,16]). In addition, it impacts the long-term prognosis, as each 5% decrease in the LVEF increases the rate of cardiac adverse events [3,17]. Even asymptomatic NYHA I rTOF individuals demonstrate a diminished LV mass–volume ratio, higher extracellular volume, systolic and diastolic LV dysfunction and thus should be actively screened for in longstanding cases [2]. The pathogenesis of LV deterioration is potentially attributable to multiple factors: [18] late or palliative shunt repair with chronic LV volume overload, long-standing cyanosis, hypoxic intraoperative injury, acquired coronary disease, dyssynchrony, etc. Hence, distinguishing between the causes of LV deterioration in rTOF patients becomes crucial for treatment decisions. Firstly, the *relatively late surgical repair* (at age 4) may explain LV dysfunction in certain adult rTOF individuals. However, it is not consistent with new-onset LV systolic deterioration and was therefore not considered causative. *Coronary artery disease* was excluded by the CTPA. *Recurrent electrical shocks* can also induce progressive myocardial dysfunction and incident heart failure episodes [19]. As previously mentioned, *dyssynchrony* may aggravate RV function (even more than PR) and LV contractility and exercise capacity [20,21]. Considering general European recommendations, the rationale and benefit of cardiac resynchronization (CRT) may arise in the presence of RBBB > 150 msec (class IIa indication) or (a weaker class IIb indication) in those with RBBB of 130 to 150 msec with RBBB ventricular activation with LVEF ≤ 35% [22]. This has also been specifically studied in rTOF patients, albeit in non-randomized designs [23]. However, CRT was not indicated for this patient since his LVEF was 48%. It is also critical to ensure prior optimal medical therapy and the tailored treatment of specific alternative causes (in this case, the newly diagnosed high PVC burden). With further regard to dyssynchrony, there is a growing body of evidence and insight regarding the importance of RBBB-mediated RV dysfunction specifically in rTOF patients. CMR studies have shown that RBBB induces the most pronounced delay in the free wall RV outflow tract subpulmonary region in rTOF patients [21]. Consequently, the potential of acute and long-term improvements in the RV function by “RV-CRT” has been sought for and demonstrated [24,25,26,27]. This concept implies atrial-tracked ventricular pacing at the site of latest RV free wall activation via dual-chamber pacemakers, allowing fusion with intrinsic physiological left-sided activation [24,25]. However, the currently existing data have not yet translated into guideline rTOF-specific indications. We seek to reevaluate the potential benefit for the RV remodeling of this intervention after PVR (class I indication for this case due to symptomatic episodes of congestion) [8]. In this particular case, the *high PVC burden*, the lack of extensive CMR LGE, the previously normal LV systolic function, the wide PVC QRS and the post-ablation LVEF recovery advocated for PVC-mediated systolic dysfunction as a probable contributing (or causative) factor to the LV systolic dysfunction. It is widely known that PVC burdens higher than 10% may lead to PVC-induced or PVC-aggravated cardiomyopathy [28,29,30]. This has also translated to a class IIa indication for ablation for high-burden PVCs which aggravate preexisting structural heart disease in recent European guidelines [31]. However, the incidence of PVC-mediated systolic dysfunction in rTOF patients remains unknown.

This case demonstrates why clinical reasoning is important to adequately identify and target pathogenic factors in rTOF patients with ventricular arrhythmia and progressive LV deterioration. Radiofrequency ablation can completely abolish VTs, which improves the LV systolic performance and long-term prognosis.

## Figures and Tables

**Figure 1 diagnostics-14-00158-f001:**
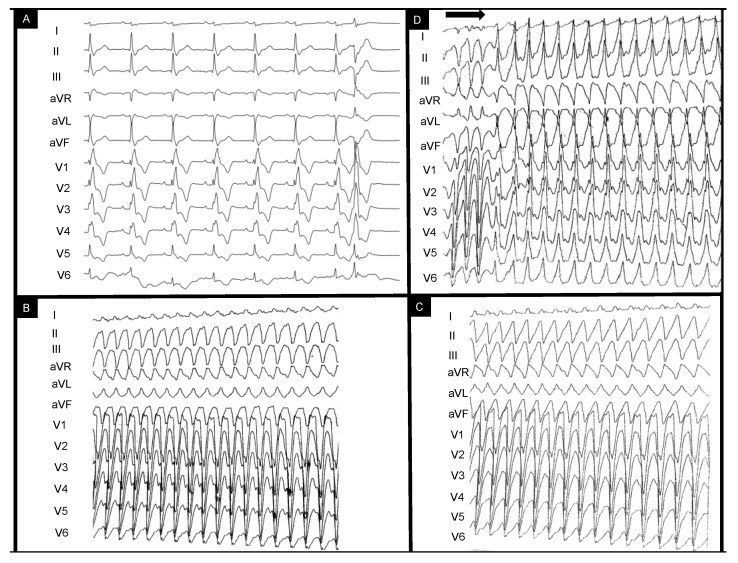
EKG-based clinical vignette regarding baseline EKG QRS morphology during sinus rhythm (**A**) and previously documented ventricular tachycardia morphologies (**B**–**D**). The horizontal arrow indicates the end-procedural programmed ventricular stimulation protocol leading to ventricular tachycardia induction. The clinical context and interpretation of the displayed tracings are further detailed below. EKG = electrocardiogram.

**Figure 2 diagnostics-14-00158-f002:**
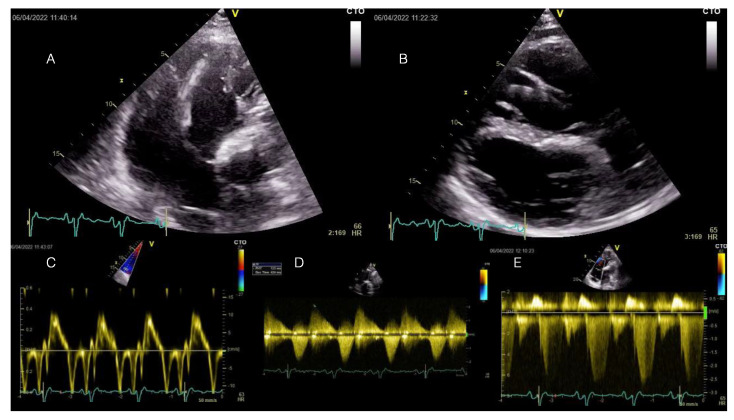
Transthoracic echocardiography examination at current admission displaying RV enlargement and moderate-to-severe tricuspid regurgitation induced by severe pulmonary regurgitation at long-term follow-up after transannular patch-based surgical repair of TOF. All displayed images are interpreted in the patient’s clinical context below. TOF = Tetralogy of Fallot.

**Figure 3 diagnostics-14-00158-f003:**
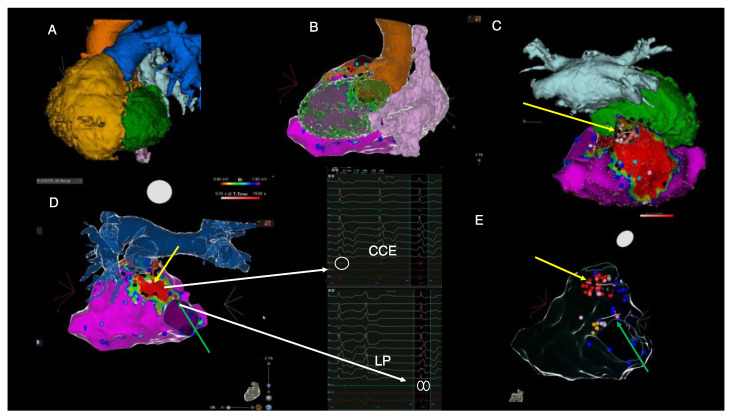
Imaging-derived clinical vignette based on preprocedural cardiac CT and intraprocedural electroanatomical mapping during the electrophysiological study. (**A**)—cardiac CT-derived left anterior oblique view of severely dilated RV (yellow), posterior position of the LV (green), pulmonary artery (blue), aorta (orange), left atrium (light blue). (**B**)—Posterolateral view of cardiac CT-derived anatomy of the left ventricle (green), right atrium (pink), aorta (orange) and electroanatomical RV voltage mapping derived from CARTO system. (**C**)—cranial view of the right ventricular electroanatomical voltage map displaying dense scar in the area of the transannular patch and the position of the left ventricle (green) and left atrium (light blue) derived from cardiac CT. (**D**)—posterior view of the right ventricular electroanatomical map fused with the cardiac CT-derived anatomy of pulmonary artery. (**E**)—posterior view of the transparent electroanatomical voltage mapping which shows final ablation sites (red dots), areas of late potentials (blue dots) and the position of His bundle (orange dots). The yellow arrow highlights the pulmonary annulus-to-septal patch anatomical isthmus, whereas the green arrow indicates the isthmus located between the septal patch and the tricuspid annulus. Red areas on voltage mapping signify dense scar tissue with low myocardial voltage (<0.5 mV). Purple areas show normal myocardial voltage < 1.5 mV). Black dots indicate the local presence of EGM signals, suggesting areas creating entry points to scar conduction channels (“CCE”). Blue dots indicate the presence of EGMs suggestive of late potentials (“LPs”). CT = computed tomography, LV = left ventricle, RV = right ventricle. These images are further discussed and interpreted below.

## Data Availability

Any data contained in this paper may be requested from the corresponding author.

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
