# Peer review of "Complex Substrate Leading to PVC-Mediated Systolic Dysfunction in addition to Sustained Monomorphic VT in Repaired Tetralogy of Fallot"

_diagnostics, 2024, doi:10.3390/diagnostics14020158_

Round 1
Reviewer 1 Report
Comments and Suggestions for Authors
I have thoroughly reviewed the article titled "Complex Substrate Leading to PVC-Mediated Systolic Dysfunction Besides Sustaining Monomorphic VT in Repaired Tetralogy of Fallot" with great interest. The case study details a patient with successfully operated Tetralogy of Fallot, who subsequently developed frequent PVCs with RBBB morphology and a superior axis (LV origin), along with ventricular dysfunction and episodes of ventricular tachycardia (LBBB morphology and superior axis).
An anatomical ablation of right ventricular isthmuses, generated during the Fallot surgery, was performed. Unexpectedly, this intervention not only addressed the induced VT but also eliminated the frequent ventricular extrasystoles that had initially suggested a left ventricular origin. The authors attribute this unexpected response to the anatomical displacement of the infundibular septum and the clockwise activation of the anatomical isthmus between the pulmonic valve and the septal patch. This results in an activation pattern with a clear morphology of right bundle branch block, suggesting a left origin, despite accessibility from the right ventricle. Subsequent to the procedure, the patient's clinical course improved significantly, with a normalization of the left ventricular ejection fraction. In the absence of alternative explanations, this improvement is tentatively attributed to possible tachycardia-induced cardiomyopathy resulting from frequent extrasystoles.
Additionally, the authors provide a comprehensive review of ventricular tachycardia mechanisms in Tetralogy of Fallot, ablation indications, and patient outcomes. While the article is well-written and engaging, I would like to suggest some modifications before publication:
1. The quality of the ECG images in Figure 1 is suboptimal. To meet publication standards, obtaining higher-resolution records with proper framing, particularly for panels 1B and 1D, is recommended.
2. The description and images of the ablation procedure require improvement. The meaning of the points of different colors projected on the map is unclear, and there is a lack of distinction between the sites of ablation and the achievement of extrasystole abolition.
I trust that addressing these suggestions will enhance the overall quality and clarity of the article.
Author Response
Dear Sir/Madam,
We have attached our letter of response in the .pdf document. Please see the attachment.
We wish to thank you for your observations and remarks and we have provided point-by-point responses to all of them. All these changes have been included in the main manuscript with tracked changes in order to improve legibility. In addition to the modifications made in response to your observations, we have also included all changes suggested by the reviewer(s).
We hope to have improved the content of this manuscript after this revision in accordance to all of your recommendations.
We thank you respectfully for this process.
On behalf of all the authors,
Dr. Cosmin Cojocaru, MD, PhD candidate
Assoc. Prof. Dr. Radu Vatasescu, MD, PhD

Reviewer 2 Report
Comments and Suggestions for Authors
The case is educative and is well documented and presented by the authors. Just a coupple of remarks, questions:
1. the possibilities of antiarrhythmic treatment in this case?
2. evidence-based treatment of HFmrEF?
3. the figures`s legends are overlapped with the text, please correct it
3. some subtitles in the text would be welcome for better understanding the presentation
4. English spelling has to be improved
Comments on the Quality of English LanguageEnglish spelling has to be improved.
Some expressions have to be changed.
Author Response

(The authors gave the same response as above.)
